# Optimal timing of a colonoscopy screening schedule depends on adenoma detection, adenoma risk, adherence to screening and the screening objective: A microsimulation study

Viktor Zaika[1,2], Meher K. Prakash[3], Chih-Yuan Cheng[4,5], Michael Schlander[4,5], Brian M. Lang[6,7], Niko Beerenwinkel[6,7], Amnon Sonnenberg[8], Niklas Krupka[2], Benjamin Misselwitz[2‡*], Jan Poleszczuk[9,10‡*]

1 Faculty of Medicine, Graduate School for Cellular and Biomedical Sciences, University of Bern, Bern, Switzerland, 2 Department of Visceral Surgery and Medicine, Inselspital Bern and Bern University, Bern, Switzerland, 3 Theoretical Sciences Unit, Jawaharlal Nehru Center for Advanced Scientific Research, Jakkur, Bangalore, India, 4 Division of Health Economics, German Cancer Research Center (DKFZ), Heidelberg, Germany, 5 Medical Faculty Mannheim, Heidelberg University, Mannheim, Germany, 6 Department of Biosystems Science and Engineering, ETH Zurich, Basel, Switzerland, 7 SIB Swiss Institute of Bioinformatics, Basel, Switzerland, 8 The Portland VA Medical Center, P3-GI, Portland, Oregon, United States of America, 9 Nalecz Institute of Biocybernetics and Biomedical Engineering, Polish Academy of Sciences, Warsaw, Poland, 10 Department of Computational Oncology, Maria Skłodowska-Curie Institute-Oncology Center, Warsaw, Poland

‡ BM and JP are contributed equally and share the last authorship
* benjamin.misselwitz@insel.ch (BM); jpoleszczuk@ibib.waw.pl (JP)

## Abstract

Colonoscopy-based screening provides protection against colorectal cancer (CRC), but the optimal starting age and time intervals of screening colonoscopies are unknown. We aimed to determine an optimal screening schedule for the US population and its dependencies on the objective of screening (life years gained or incidence, mortality, or cost reduction) and the setting in which screening is performed. We used our established open-source microsimulation model CMOST to calculate optimized colonoscopy schedules with one, two, three or four screening colonoscopies between 20 and 90 years of age. A single screening colonoscopy was most effective in reducing life years lost from CRC when performed at 55 years of age. Two, three and four screening colonoscopy schedules saved a maximum number of life years when performed between 49–64 years; 44–69 years; and 40–72 years; respectively. However, for maximum incidence and mortality reduction, screening colonoscopies needed to be scheduled 4–8 years later in life. The optimum was also influenced by adenoma detection efficiency with lower values for these parameters favoring a later starting age of screening. Low adherence to screening consistently favored a later start and an earlier end of screening. In a personalized approach, optimal screening would start earlier for high-risk patients and later for low-risk individuals. In conclusion, our microsimulation-based approach supports colonoscopy screening schedule between 45 and 75 years of age but the precise timing depends on the objective of screening, as well as assumptions regarding

**Data Availability Statement:** All relevant data are within the manuscript and its Supporting information files.

**Funding:** This work was funded by the Swiss Cancer League, grant No. KFS-5164-08-2020 to Benjamin Misselwitz. The funders had no role in study design, data collection and analysis, decision to publish or preparation of the manuscript.

**Competing interests:** The authors have declared that no competing interests exist.

**Abbreviations:** ADR, Adenoma detection rate; CMOST, Colon Modeling Open Source Tool; CRC, Colorectal cancer; CRC-SPIN, Colorectal Cancer Simulated Population model for Incidence and Natural history; ICER, Incremental Cost Effectiveness Ratio; LYG, Life years gained; MISCAN, Microsimulation Screening Analysis; SimCRC, Simulation Model of Colorectal Cancer (SimCRC); USPTF, US Preventive Services Task Force.

individual CRC risk, efficiency of adenoma detection during colonoscopy and adherence to screening.

## Introduction

Colorectal cancer (CRC) is an important public health burden in industrialized countries and the developing world [1,2]. CRC ranks third and second with respect to incidence and mortality among all cancers in the United States and in many other industrialized countries [3,4]. CRC screening tests based on the detection of occult blood in the stool [5–12] or on endoscopic visualization of the colon [13–20] have proven effective in reducing mortality. Screening generally detects CRC at earlier stages with better chances of survival. The fecal immune test (FIT) has been widely used for CRC screening, providing a highly cost-effective option for the prevention of CRC-related mortality in asymptomatic individuals of average risk [21–23].

Endoscopy is another attractive option for CRC screening with the potential to prevent CRC since adenomatous precursors of CRC can be removed during endoscopy. The effectiveness of rectosigmoidoscopy in reducing CRC incidence and mortality was demonstrated in several large randomized controlled studies [13,15,19,24]. Colonoscopy may be similarly effective as rectosigmoidoscopy [16,25–29], but rigorous randomized controlled trials have been initiated only recently [29–32]. CRC screening has been introduced in several industrialized countries and in most settings, a choice of FIT, rectosigmoidoscopy and/ or colonoscopy is offered.

Screening by FIT or colonoscopy is generally recommended for asymptomatic individuals between 45 (or 50) and 75 years of age at 1- or 2-year intervals for FIT and 10-year intervals for colonoscopy [33–35]. However, starting of screening at age 45 instead of 50 years is controversial [36–39]. Given the high public health burden of CRC, validating, and optimizing screening schedules remains a central task of gastroenterology. This is especially relevant in countries or settings when resources for CRC screening are limited [40]. Because of the complexity of CRC screening and the inherent limitations of randomized clinical trials, many questions regarding the best mode of CRC prevention are unlikely to be answered by clinical studies alone, thus requiring incorporating modelling-based solutions [41–43].

Microsimulation is a powerful method for modeling the natural history of CRC and screening interventions [44]. The natural history and the impact of screening are simulated in a large population of individual patients. Various microsimulation models including the Microsimulation Screening Analysis (MISCAN), Colorectal Cancer Simulated Population model for Incidence and Natural history (CRC-SPIN) and Simulation Model of Colorectal Cancer (SimCRC) and newer models such as CRC-AIM have been used to evaluate different types of screening protocols [44–47]. The outcome predictions of individual models regarding the effectiveness of screening vary [48,49].

Recent microsimulation studies by the US Preventive Services Task Force (USPSTF) compared the effectiveness of possible screening strategies, which varied regarding screening method and timing [50,51]. From a total of over one hundred strategies including 20 colonoscopy-based strategies, race- and age-based recommendations were made [51]. For ease of use and interpretation, only screening timepoints differing by 5- or 10-year increments were tested. Therefore, an unprejudiced search of the whole parameter space for an optimal timing of multiple screening colonoscopies has not been performed. Further, MISCAN, SimCRC, and CRC-SPIN are proprietary, thus calculations cannot be independently replicated or advanced.

In the present study, we use the open-source microsimulation framework *Colon Modeling Open Source Tool* (CMOST), calibrated to the US population [52] to search for an optimal

screening schedule with 1–4 screening colonoscopies between the ages of 20 to 90 years regarding reduction in CRC incidence and mortality as well as life years gained (LYG) and total costs. Results were subject to a rigorous sensitivity analysis regarding key parameters such as the adenoma rate, the adenoma detection rate, and adherence to screening.

## Methods

### Microsimulation model

For our calculations, we used our previously described open-source framework Colon Modeling Open Source Tool (CMOST) [52]. CMOST has been calibrated to reflect the natural history of CRC in the US population (see S1 Table). We re-implemented the CMOST Tool in C++ to enhance computational efficiency without altering the flow and the logic of the simulation or time increments for calculations (3 months). This adaptation maintains the integrity of the original model while significantly improving processing speed, enabling more extensive and complex analyses within shorter timeframes. In contrast to the original model, CMOST now uses SEER CRC incidence and mortality data from the years 1988 to 2002 [42,53], before the onset of wide-spread CRC screening (S1 Table). Additional basic parameters for CMOST calculations are provided in S2 Table. CMOST had been validated against the outcome of the existing microsimulations models MISCAN, SimCRC and CRCSpin as well as an CRC prevention study [52].

The new code is available at: https://doi.org/10.5281/zenodo.4122105.

The model structure of CMOST has been described in detail [52]. In brief, in CMOST, carcinoma develops via early and advanced adenoma precursors. Altogether six adenoma stages and four carcinoma stages are considered in the model. Most of the transition towards pre-clinical cancer occurs via the adenomatous pathway, advancing through the six successive stages of adenoma progression. With lower likelihoods, pre-clinical cancer can also start from any of the adenoma stages or even from seemingly normal colon mucosa. This "direct cancer path" represents difficult-to-prevent CRC either emerging without adenomatous precursors [54,55], e.g. in connection with CTTNB1 mutations, deriving from exceedingly hard-to-detect adenomas also missed by tandem-colonoscopies [56] or very fast-growing adenomas with a dwell time < 3 years.

CMOST accounts for the gender- and age-dependent risks for adenoma development [26,57], the presence of multiple adenomas, as well as their locations within the colon. Colonoscopy is performed according to a pre-defined screening schedule or for symptomatic cancer. The model accounts for varying probabilities of visualizing the hepatic and splenic flexures, adenoma detection [58,59], and complications of colonoscopy [60]. We also simulate surveillance colonoscopies after the detection of a lesion according to the US Multi-Society Task Force on Colorectal Cancer guidelines [61]. Detailed information about default model settings and parameters can be found in the code repository.

Calibration overall worked reasonably well, except at the ages below 40 years when the prevalence of adenomas or cancer is low (S1 Table). Specifically, 56%, 100% and 81% of the calculated values for early, and advanced adenoma and cancer, respectively in individuals >40 years were within 20% of the benchmark. However, we note a lower adenoma and carcinoma incidence between the ages of 40–50 years than expected in the calibrated model. Further, regarding adenoma stage, the fraction of very small adenomas was slightly lower than in the benchmarks and the fraction of individuals with 3, 4, and 5 adenomas was also lower than expected. Moreover, our model calculated slightly (<20%) more advanced cancers (stages III +IV) then benchmarked. Consequences of these reported discrepancies between calibration and benchmarks regarding our study questions are difficult to predict. The lower adenoma

and CRC incidence at age 45 might favor screening at later time points. However, the remaining discrepancies are unlikely to change our estimation of colonoscopy timing in a relevant manner.

CMOST microsimulation tracks the history of a general population in the US from birth until death for a maximum age of 100 years. Adenoma initiation, progression to advanced adenoma and cancer, cancer progression, screening and surveillance are all modeled in time increments of 3 months. Death results from CRC, medical interventions, or from other age-dependent causes of mortality at rates according to 2008 US Life Table Data [62]. Death from colorectal cancer is restricted to the first five years after the initiation of cancer according to published survival rates [63]. An individual patient who survives CRC is followed until death from other causes. Several measures of effectiveness are calculated, such as LYG per 1000 individuals (i.e. the reduction in life-years lost to CRC death and CRC screening procedures), as well as reduction in CRC-associated incidence and mortality.

## Simulation and optimization

Due to the stochastic nature of the model, we performed 10 independent simulations for each set of evaluated model parameters, all using a large population of 20 million individuals. We use the Mercenne-Twister uniform pseudo-random number generator, to reproducibly create pseudonumbers for the 10 runs of the simulation. The results are quite consistent for simple outcomes such as incidence reduction due to colonoscopy screening. However, relevant fluctuations of results were observed for complex calculations such as the sensitivity analysis. For these results we illustrate the precision of our calculations by providing margins of errors (interquartile range, $10^{th}$ and $90^{th}$ percentile and outliers). The Wilcoxon rank-sum test was used to compare simulation outputs. Simulations were carried out on a dedicated computational cluster consisting of 192 CPUs and 312 GB of random-access memory.

Search for the optimal colonoscopy schedule was always performed for a given set of model parameters, with the search space defined by the minimum and maximum possible screening ages (20 and 90 years, respectively), number of screening colonoscopies (up to four), and minimum time separation between colonoscopies (5 years). For simplicity, we assumed that the search space consists only of integer screening ages. During the optimization for any input value from the search space that was requested by an optimization procedure we have run the whole model simulation and then evaluated each of four considered model outputs: 1) reduction of mortality (in %), 2) reduction of incidence (in %), 3) reduction of discounted life years lost (in %), and 4) reduction of total treatment costs (in %); all reductions in comparison to the no screening scenario. We looked for optima for each of those outputs separately as well as for non-dominated or Pareto optimal screening schedules, for which no function value can be improved without degrading at least one other objective value.

In the case of scenarios with one and two colonoscopies we used the brute force approach, i.e. we run simulations for all possible points in the search space, both when looking for optimal scenarios for the nominal set of parameters and when performing sensitivity analyses. Non-dominated solutions were found by using Kung's algorithm [64] on all evaluated schedules. For three and four colonoscopies, due to the increased size of the search space, we used several hundred iterations of the Mixed Integer Distributed Ant Colony Optimizer (MIDACO) solver [64] in order to obtain good estimates for the optima and non-dominated solutions. MIDACO is a meta-heuristic multi-objective optimization solver which allows to look at all objectives simultaneously and produces a set of non-dominated solutions (optimal solutions for a given output are within this set) but can't guarantee optimality as any other non-deterministic algorithm. Therefore, when presenting the results we have also plotted the

landscape of the optimization problem to see if the resulting screening schedules are withing compact regions.

## Costs and cost-effectiveness

Costs were considered from a healthcare payer's perspective. Cost assumptions for the United States were taken from a recent study [65] and transformed to 2020 United States dollars [66]. Our model includes costs for screening colonoscopy, polypectomy, procedural complications, and CRC treatment (see S2 Table). For all calculations, costs are computed for 3 months intervals. Costs differ according to time after diagnosis and we distinguish three different time periods: the first 12 months following diagnosis, the last 12 months before CRC-related death and the time interval in between (follow-up period). To avoid overlap of these time periods in case of a short time interval between diagnosis and death, for each 3-months increment, only one type of costs is considered with costs for the last 12 months taking precedence over costs for the first 12 months and the follow-up period. In case of death unrelated to CRC, additional costs apply as described [65]. CRC-related costs differ according to CRC stage and were considered for up to 5 years after diagnosis.

Screening schedules with one and two screening colonoscopies were cost saving and an absolute optimal solution with minimal costs was observed inside the assumed range from 20 to 90 years. In contrast, for schedules with 3 and 4 colonoscopies, no optimal timing was found: optimization for costs yielded a scheduling of the third and fourth interventions very late in life (i.e. at 90 years, the upper limit of the assumed range). This results in non-participation of a large fraction of the population, which resulted in lower costs.

In line with current practice in health economics, for cost-effectiveness calculations, life years gained as well as CRC screening and treatment costs were discounted. We used a 3% discounting rate per year (consumption rate) [67], starting from age 20, resulting in discounted US dollars (dUSD) and discounted life years gained (dLYG). For calculation of incremental cost-effectiveness ratios (ICERs) we considered all screening schedules that saved more discounted life years than "standard screening" schedule at ages 50, 60, and 70. We considered all ICER values below 100'000 dUSD/ dLYG as cost-effective [68,69].

## Sensitivity analysis

For sensitivity analysis of optimal schedules, five key parameters were selected and used in a one-way sensitivity analysis: individual adenoma risk, adenoma detection rate, adherence to screening, survival rates after CRC diagnosis, and frequency of complications. Variations in adenoma risk could help adapting our calculations to other countries with a different CRC incidence compared to the US. For all parameters, baseline values were increased and decreased by 10%, 25%, and 50%, followed by another search for an optimum. Baseline assumptions are provided in S2 Table (e.g., 100% for adherence to colonoscopy. Assumptions were bounded at 100% for adenoma detection, adherence, and CRC survival. In addition to this systematic parameter variation, we also estimated upper and lower bounds of some parameters using real-life data: i) *CRC survival*: Future medical breakthroughs may improve CRC treatment. For melanoma treatment, the use of checkpoint inhibitors is associated with a hazard ratio (HR) of dying of 0.39 [70]. Vice versa, an approximately twofold higher mortality was observed in some countries (e.g. Poland) compared to the group with the best outcomes [71]. We therefore varied CRC mortality by a HR of 0.39 and 2 compared to the standard scenario. ii) *Complications*: the sum of serious complications (perforation, bleeding, serosa burn) was varied between 0.1% and 1.5%[72]. iii) *Cecal intubation rate* was as low as 70% in some studies [58,73,74].

### Personalized screening

For personalization of CRC screening, we simulated a hypothetical test of individual adenoma risk, which stratifies the population into a high-risk (25% of individuals) and a low-risk group. Our approach simulated a hypothetical, imperfect test with individuals in the low- and high-risk groups that will and will not develop cancer, respectively. The test was calibrated such that the high-risk group had an average 1.5, 2.0, 2.5, or 3.0-fold increased CRC risk compared to the low-risk group. Even though our tests is purely hypothetical and not based on actual groups within the population, a number of such tests have been described using epidemiological data (e.g. BMI, gender, nutrition, activity, smoking, family history) and endoscopy results [75], including the validated Colorectal Cancer Risk Assessment Tool by NIH [76]. We searched for an optimal schedule in the low-risk and in the high-risk population for different risk ratios of the test.

## Results

### Identification of optimal screening schedules

We used our open-source microsimulation tool CMOSTv3 to simulate the natural history of CRC and colonoscopy-based CRC screening in the US population. We tested scheduling of 1–4 screening colonoscopies between the ages 20 and 90. For each strategy, we calculated an optimal schedule, minimizing either: i) CRC incidence, ii) CRC mortality, iii) life years lost to CRC, or iv) overall costs. No single optimal schedule for all outcomes could be identified. We therefore identified screening schedules that cannot be further optimized without compromising on at least one of the four considered outcomes (Pareto-optimal solutions).

### Single screening colonoscopy

We first assumed a healthcare setting with limited resources, which allows only one screening colonoscopy. In this setting, incidence reduction was optimal at 61 years, mortality reduction at 63 years, and cost reduction was optimal at 61 years (Table 1). However, most life years were saved when colonoscopy was performed considerably earlier in life, at 55 years (Fig 1A). Considering having four optimization goals we identified 9 Pareto solutions ("compromises"), for which further optimizing of one outcome would compromise on another.

In our sensitivity analysis, the screening schedule for the maximum number of LYG was sensitive to adenoma detection during colonoscopy and baseline adenoma risk (Fig 1B). The optimal time point was later in life under the assumption of less efficient adenoma detection rates during colonoscopy. Similarly, a lower individual adenoma risk also favored a later start of screening. However, the median screening years only varied by 2–3 years, upon increase or reduction of the model parameter of interest by 50% (Fig 1B). For a single colonoscopy program, non-adherence to the single colonoscopy would simply result in non-participation in the screening program. Therefore, adherence did not affect the optimal time point of colonoscopy in our sensitivity analysis.

We also tested the sensitivity of the screening schedule for optimal incidence, mortality, and cost reduction, respectively (Figs 1B and S1). Lower baseline adenoma risk consistently resulted in a later start of screening. Interestingly, in a setting of decreased cancer survival, and earlier start of cancer screening would result in lower cancer mortality. In contrast, variations in colonoscopy complication rates or cecal intubation rates (not shown) did not affect the screening schedule in any scenario.

**Table 1. Results for optimal screening strategies in the US population.**

| | Year (s) | Incidence reduction | Mortality reduction | Life years gained# | Costs | Cost-effectiveness (dUSD/dLYG) | Number colonoscopies per 1000 | Colonoscopies per LYG | Colonoscopies per case prevented |
|---|---|---|---|---|---|---|---|---|---|
| **One screening colonoscopy** | | | | | | | | | |
| **Life years gained** | **55** | **37.5% ± 0.04%** | **40.8% ± 0.08%** | **40.6% ± 0.1% (114 ± 0.2 years)** | **20% ± 0.07%** | **-6175.1 ± 69.78** | **1487.2 ± 0.38** | **13 ± 0.03** | **58.3 ± 0.1** |
| Incidence reduction | 61 | 39.8% ± 0.04% | 44.6% ± 0.1% | 37.7% ± 0.13% (105.9 ± 0.3 years) | 22.3% ± 0.06% | -10022.3 ± 61.58 | 1460.3 ± 0.47 | 13.8 ± 0.04 | 53.9 ± 0.08 |
| Mortality reduction | 63 | 39.4% ± 0.05% | 44.7% ± 0.05% | 35.5% ± 0.08% (99.6 ± 0.3 years) | 22% ± 0.07% | -10991.4 ± 88.21 | 1436.2 ± 0.42 | 14.4 ± 0.04 | 53.7 ± 0.09 |
| Total costs | 61 | 39.8% ± 0.04% | 44.6% ± 0.1% | 37.7% ± 0.13% (105.9 ± 0.3 years) | 22.3% ± 0.06% | -10022.3 ± 61.58 | 1460.3 ± 0.47 | 13.8 ± 0.04 | 53.9 ± 0.08 |
| Balanced solution | 59 | 39.7% ± 0.05% | 44% ± 0.08% | 39.5% ± 0.09% (111.1 ± 0.2 years) | 22% ± 0.07% | -8845.1 ± 84.14 | 1476.2 ± 0.46 | 13.3 ± 0.03 | 54.8 ± 0.11 |
| **Two screening colonoscopies** | | | | | | | | | |
| **Life years gained** | **49, 64** | **50.4% ± 0.03%** | **55.2% ± 0.06%** | **53.7% ± 0.1% (150.9 ± 0.3 years)** | **17.6% ± 0.06%** | **593.8 ± 48.85** | **2490 ± 0.49** | **16.5 ± 0.03** | **72.7 ± 0.1** |
| Incidence reduction | 53, 69 | 52.1% ± 0.05% | 58.3% ± 0.07% | 52.2% ± 0.09% (146.6 ± 0.4 years) | 20% ± 0.08% | -1963.1 ± 59.93 | 2411.5 ± 0.36 | 16.5 ± 0.04 | 68.1 ± 0.11 |
| Mortality reduction | 55, 71 | 51.9% ± 0.05% | 58.5% ± 0.07% | 50.4% ± 0.1% (141.5 ± 0.5 years) | 20.6% ± 0.07% | -3315.6 ± 51.21 | 2358.9 ± 0.47 | 16.7 ± 0.05 | 66.9 ± 0.09 |
| Total costs | 57, 73 | 51.1% ± 0.04% | 58.2% ± 0.08% | 48.2% ± 0.14% (135.4 ± 0.4 years) | 20.7% ± 0.06% | -4485.1 ± 55.47 | 2293.1 ± 0.44 | 16.9 ± 0.05 | 66 ± 0.09 |
| Balanced solution | 53, 69 | 52.1% ± 0.05% | 58.3% ± 0.07% | 52.2% ± 0.09% (146.6 ± 0.4 years) | 20% ± 0.08% | -1963.1 ± 59.93 | 2411.5 ± 0.36 | 16.5 ± 0.04 | 68.1 ± 0.11 |
| **Three screening colonoscopies** | | | | | | | | | |
| **Life years gained** | **44, 57, 69** | **56.3% ± 0.04%** | **62.2% ± 0.1%** | **60.2% ± 0.07% (169.1 ± 0.3 years)** | **10.4% ± 0.08%** | **6497.3 ± 48.43** | **3374.2 ± 0.35** | **20 ± 0.03** | **88.2 ± 0.09** |
| Incidence reduction | 48, 63, 73 | 57.6% ± 0.04% | 64.3% ± 0.07% | 58.7% ± 0.11% (164.8 ± 0.3 years) | 13.4% ± 0.08% | 3564.8 ± 52.41 | 3233.7 ± 0.47 | 19.6 ± 0.04 | 82.6 ± 0.1 |
| Mortality reduction | 50, 65, 76 | 57.4% ± 0.04% | 64.8% ± 0.09% | 57.4% ± 0.13% (161.3 ± 0.4 years) | 14.6% ± 0.06% | 2448.8 ± 55.32 | 3128.2 ± 0.47 | 19.4 ± 0.05 | 80.2 ± 0.08 |
| Balanced solution | 49, 64, 75 | 57.5% ± 0.04% | 64.7% ± 0.09% | 58% ± 0.09% (163 ± 0.5 years) | 14.2% ± 0.08% | 2886.9 ± 57.66 | 3173.5 ± 0.51 | 19.5 ± 0.05 | 81.2 ± 0.08 |
| "standard screening" | 50, 60, 70 | 56.8% ± 0.04% | 63% ± 0.08% | 58.9% ± 0.12% (165.5 ± 0.4 years) | 12% ± 0.09% | 5262.3 ± 62.82 | 3291.6 ± 0.39 | 19.9 ± 0.05 | 85.2 ± 0.12 |
| **Four screening colonoscopies** | | | | | | | | | |
| **Life years gained** | **40, 51, 61, 72** | **59.8% ± 0.04%** | **66.1% ± 0.09%** | **63.9% ± 0.08% (179.4 ± 0.4 years)** | **1% ± 0.1%** | **13831.7 ± 65.03** | **4250.7 ± 0.36** | **23.7 ± 0.05** | **104.6 ± 0.12** |
| Incidence reduction | 45, 57, 67, 77 | 60.9% ± 0.03% | 68.4% ± 0.1% | 62.9% ± 0.13% (176.7 ± 0.3 years) | 5.4% ± 0.08% | 8919.1 ± 45.35 | 4004.3 ± 0.47 | 22.7 ± 0.04 | 96.7 ± 0.12 |

*(Continued)*

**Table 1.** (Continued)

| Mortality reduction | 47, 59, 69, 79 | 60.7% ± 0.03% | 69% ± 0.06% | 62.3% ± 0.1% (175 ± 0.2 years) | 6.8% ± 0.09% | 7517.2 ± 54.17 | 3887.9 ± 0.36 | 22.2 ± 0.03 | 94.2 ± 0.13 |
| Balanced solution | 46, 59, 69, 79 | 60.7% ± 0.03% | 69% ± 0.05% | 62.4% ± 0.12% (175.3 ± 0.4 years) | 6.7% ± 0.08% | 7532.6 ± 53.32 | 3893.9 ± 0.45 | 22.2 ± 0.05 | 94.4 ± 0.12 |

## Two screening colonoscopies

Next, we considered a setting that would allow for two screening colonoscopies during the lifetime of an individual. In this scenario, optimal incidence and mortality reduction were achieved at 53 and 69 years, and at 55 and 71 years, respectively (Fig 2A, Table 1). As observed for a single colonoscopy, the maximum of life years was saved upon an earlier start of screening at 49 and 64 years. We also identified several Pareto solutions ("compromises" between outcomes, compare yellow circles in Fig 2).

As for the one-colonoscopy scenario, the optimal screening schedule was sensitive to variations in adenoma detection rates and adenoma risk. Less efficient adenoma detection and lower baseline adenoma risk yielded in a later time point of the first screening colonoscopy (Fig 2B). Interestingly, lower adherence to screening yielded in a later first and an earlier second colonoscopy, thus shifting the schedule closer to the optimal time point of a single colonoscopy. Lower adenoma detection efficiency and lower baseline adenoma rates also resulted in a later screening start for optimal incidence, mortality, and cost reduction (S2 Fig).

## Three screening colonoscopies

A number of international guidelines suggest a CRC screening schedule with 3 screening colonoscopies at 10-year intervals between 50 and 75 years of age [35,77]. When simulating a scenario with three screening colonoscopies, we found optimal incidence reduction at 48, 63, and 73 years, and optimal mortality reduction at 50, 65, and 76 years (Fig 3A, Table 1). As above, an optimal number of LYG would require earlier screening at 44, 57, and 69 years. Thus, this optimized screening schedule would somewhat outperform "standard screening" at ages 50, 60 and 70 years. As described for the 1- and 2-colonoscopy schedule, a number of Pareto solutions ("compromises") were identified.

The three-colonoscopy screening schedule showed similar dependency on assumptions as the two-colonoscopy schedule: Lower adenoma detection rates and lower individual adenoma risk favored a later start of screening (Fig 3B). Further, lowered screening participation resulted in a screening schedule with a later first and earlier third colonoscopy, approaching the time window of the two-colonoscopy screening schedule.

Similar dependencies were observed for the screening schedule for optimal incidence and mortality reduction (S3 Fig). Interestingly, for these outcomes lower adenoma detection also shifted the timing of the third colonoscopy by 2 years (from 73 to 71 for optimal incidence reduction, and from 76 to 74 years for optimal mortality reduction, respectively) resulting in a narrow time for screening.

## Four screening colonoscopies

Several US current guidelines for CRC screening suggest CRC screening between the ages of 45 to 75 years at 10-year intervals, which might result in up to four screening colonoscopies within the lifetime of an individual [33,34,36]. The addition of a fourth endoscopy further

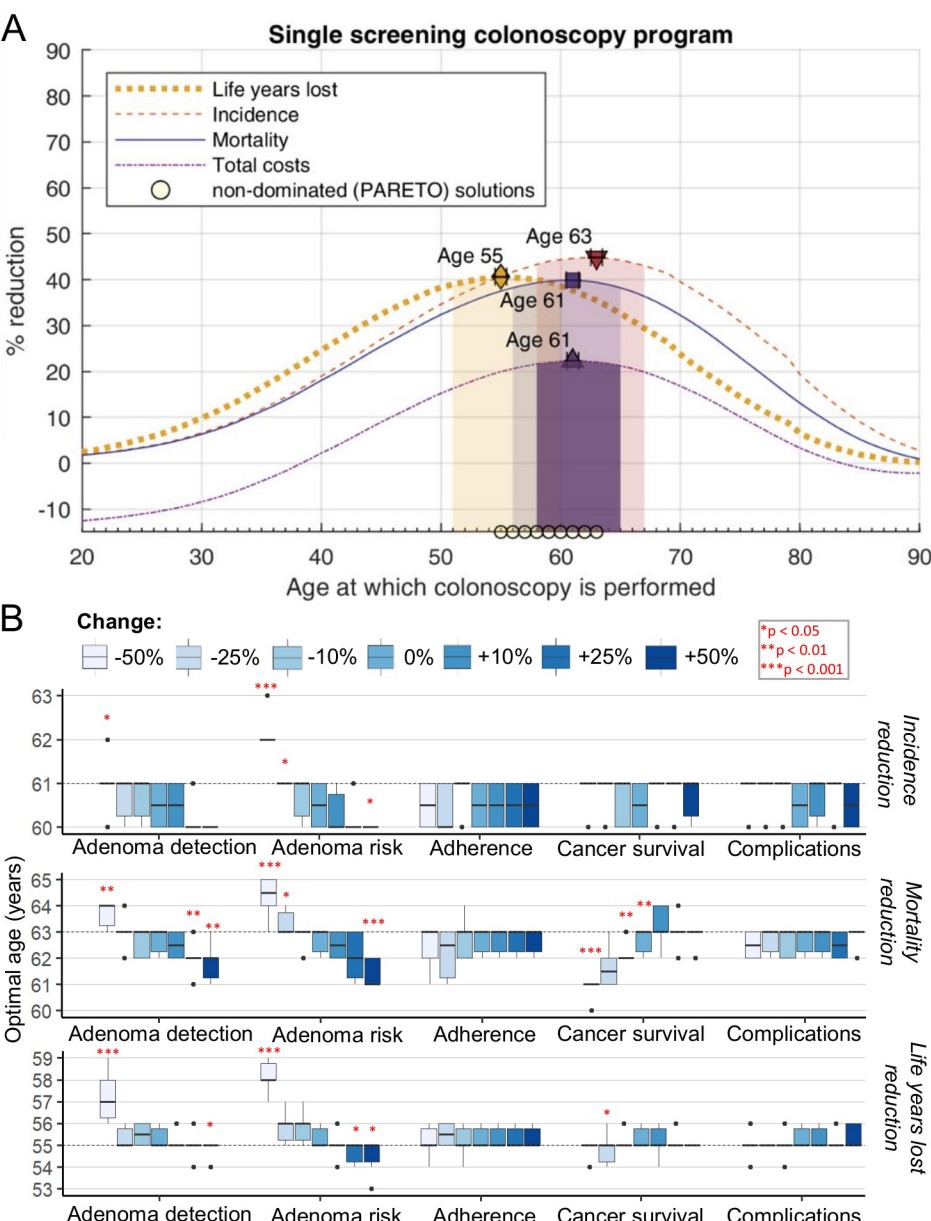

**Fig 1. Optimal screening time point for a single colonoscopy program in the US. (A)** Reduction in life years lost, CRC incidence, mortality and total costs for different time points of a single screening colonoscopy with 100% adherence. Optimal points for reduction in CRC incidence, mortality and life years lost are indicated, together with corresponding shaded regions where 95% of maximal reduction is achieved. Open circles: Pareto non-dominated solutions. **(B)** Results of sensitivity analysis of the optimal screening point for reduction of lost life years, incidence and mortality reduction. Selected model parameters were increased and decreased by up to 50%, as indicated. The dashed line indicates the optimal schedule without parameter changes. Significance is tested using the Wilcoxon test against a distribution with no change in the parameter. *$p < 0.05$, **$p < 0.01$. ***$p < 0.001$. For all calculations in both panels N = 10 simulations on a population of 20 million individuals are used, and the error of the precision of our analysis is indicated. Reading example: Upper subpanel—for a lower adenoma risk (-50%—light blue bar) the optimal year for incidence reduction for the single screening colonoscopy is 62 years, instead of 60–61 years for standard adenoma risk (p < 0.001 when comparing 10 repeat calculations). Middle subpanel: Lower adenoma risk (-50%, light blue bar) also favored later screening for optimal mortality reduction (at 64–65 years compared to 63 years with standard risk). Lower subpanel: The optimum for life years lost reduction is found at 58 years compared to 56 years with standard risk.

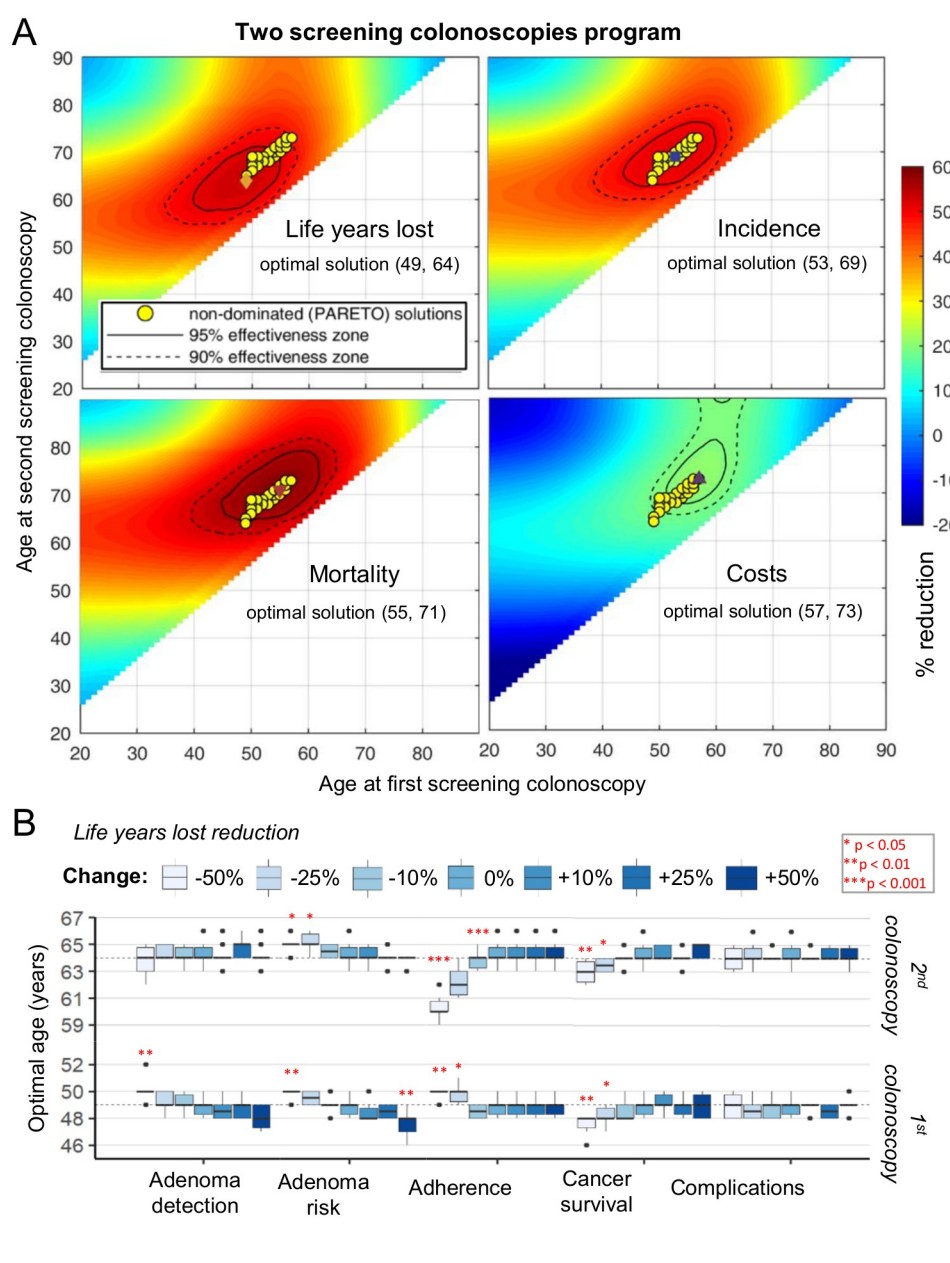

**Fig 2. Optimal time points for a two-colonoscopy program in the US. (A)** Reduction in life years lost, CRC incidence, mortality, and total costs for a two-colonoscopy screening program with 100% adherence. Optimal time points and corresponding regions where 90% (dashed line) and 95% (solid line) of maximal reduction is achieved. Yellow circles: Pareto non-dominated solutions. **(B)** Results of a sensitivity analysis for the optimal screening time points for a maximum reduction in life years lost. Selected model parameters were increased and decreased by up to 50%. Dashed lines indicate average optimal points obtained for nominal is simulations. Significance is tested using the Wilcoxon test against a distribution with no change in the parameter. $^*p < 0.05$, $^{**}p < 0.01$. $^{***}p < 0.001$. For all calculations in both panels N = 10 simulations on a population of 20 million individuals are used, and the error of the precision of our analysis is indicated. Reading example: For a lower adherence to screening (-50%—light blue) the optimal time point for the first screening colonoscopy for maximum reduction of life years lost is found at age 50 years, compared to 49 years with standard screening adherence (100%), p < 0.01. The optimal age for the second colonoscopy is found at age 60 years, compared to 64 years with standard adherence (p < 0.001).

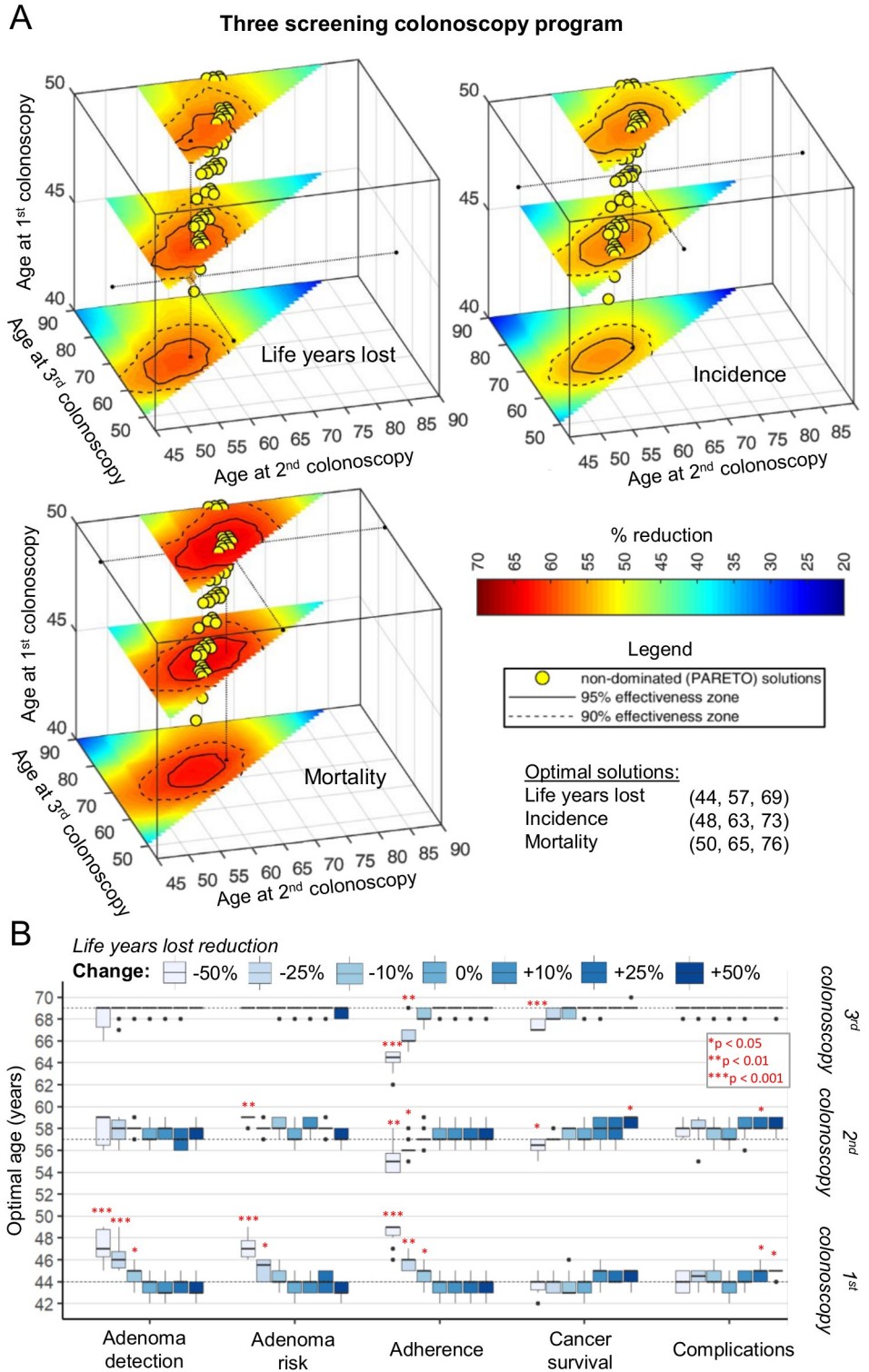

**Fig 3. Optimal time points for a three-colonoscopy screening program in the US. (A)** Reduction in life years lost, CRC incidence and mortality and total costs for a three-colonoscopy screening program with 100% adherence. Exemplary slices of the three-dimensional space of three colonoscopy time points are indicated with regions where 90% (dashed line) and 95% (solid line) of maximal reduction is achieved. Yellow circles: Pareto non-dominated solutions. **(B)** Results of a sensitivity analysis for the optimal screening time points for a maximum reduction in life years lost. Selected model parameters were increased and decreased by up to 50%. Dashed lines indicate average

optimal points obtained for nominal simulations. Significance is tested using Wilcoxon test against a distribution with no change in the parameter. $^*p < 0.05$, $^{**}p < 0.01$. $^{***}p < 0.001$. For all calculations in both panels N = 10 simulations on a population of 20 million individuals are used, and the error of the precision of our analysis is indicated. <u>Reading example</u>: For a lower adherence to screening (-50%—light blue) the optimal time point for the first screening colonoscopy for maximum reduction of life years lost is found at age 49 years, compared to 45 years with standard screening adherence (100%), p < 0.001. The optimal age for the second and third colonoscopies are found at ages 55 and 64–65 years, compared to 57 and 68 years with standard adherence (p < 0.01).

increased efficacy of screening with lower CRC incidence and mortality and a higher number of life years saved (Table 1). With four colonoscopies, screening would start before the age of 50 for all outcomes: incidence reduction was optimal at 45, 57, 67, and 77 years, and mortality reduction was optimal at 47, 59, 69, and 79 years. Maximizing the number of LYG would require a start of screening at 40, followed by screenings at age 51, 61, and 72 years.

Key outcome measures for identified screening strategies for optimal life years gained (bold), incidence and mortality reduction, costs and a balanced solution are provided. *To illustrate the precision of our analyses, values are given as average ± standard deviation of N = 10 simulations on a population of 20 million. #Life years gained comprise the reduction in life-years lost to CRC death and CRC screening procedures; the absolute number of life years gained are provided in round brackets.

The sensitivity analysis favored a later start of screening in a scenario with lower adenoma detection for incidence and mortality reduction, but no dependencies of the screening schedule regarding adenoma rates. Lower adherence favored a later start and an earlier end of screening for all outcomes, resulting in a more time window (S4 Fig).

## Effectiveness and cost-effectiveness

For calculating cost-effectiveness, life-years were discounted at 3% per year starting at age 20 years. Colonoscopy was effective for prevention of life years lost due to CRC, and more colonoscopies resulted in a higher number of LYG. Screening with four colonoscopies at 40, 51, 61, and 72 years saved the highest number of discounted life years (Fig 4). In parallel, more colonoscopies resulted in higher number of discounted USD (not shown). Using this information, we calculated incremental cost-effectiveness ratios (ICER), which represent additional

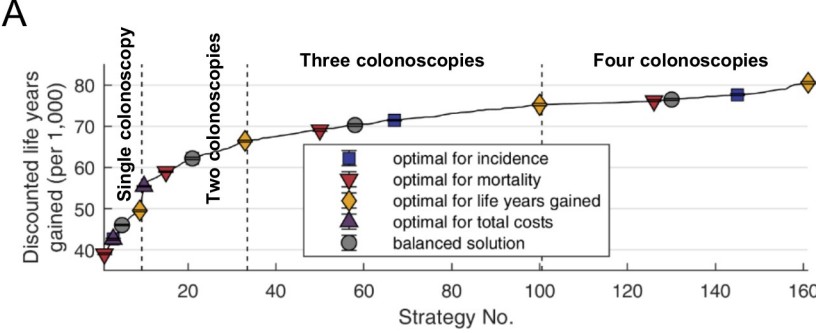

**Fig 4. Effectiveness analysis.** All 560 non-dominated Pareto solutions including optimal solutions for CRC incidence and mortality reduction, reductions in life years lost and costs were sorted to the resulting number of discounted life years gained (dLYG). 398 scenarios with N colonoscopies resulted in lower dLYG than any of the scenarios with N−1 colonoscopies and were discarded.

costs required to save an additional number of life years. For simplicity, only the 4 scenarios with optimal LYG were considered. Among these, no dominant strategy was identified.

Calculation of the ICERs for 1 colonoscopy vs. no screening revealed that this intervention was cost saving (-6'169 (± 57) dUSD/ dLYG). Screening with 2 colonoscopies (vs. 1 colonoscopy) and 3 colonoscopies (vs. 2 colonoscopies) was cost-effective with ICERs of 20'407 (± 315) dUSD/ dLYG and 51'005 ± 114) dUSD/ dLYG, respectively. In contrast, a strategy with 4 colonoscopies was not cost-effective compared to 3 colonoscopies with an ICER of 118'265 dUSD/ dLYG (± 3'346.6), above the accepted threshold of 100'000 dUSD/ dLYG.

## Personalized CRC screening

Individuals differ regarding their personal CRC risk. When we stratified the population according to adenoma risk (see method section), higher personal CRC risk favored an early start of screening while lower CRC risk favored later screening (Fig 5). Specifically, the first colonoscopy was 8, 6, 6, and 2 years earlier for 1, 2, 3, and 4 colonoscopy schedules, respectively, in individuals with the 2–3 fold higher CRC risk compared to individuals with a lower risk. The remaining screening colonoscopies were also scheduled earlier by a smaller number of years. Optimal screening schedules further differed between male and female subjects and our results suggest a start of colonoscopy screening 2–3 years earlier in men than in women (not shown).

## Discussion

In the present study, we use a microsimulation model for the natural history of CRC to optimize the schedule of colonoscopy screening in the US population. Our findings indicate that the optimal timing for screenings varies significantly depending on the targeted outcomes incidence and mortality reduction or number of LYG, respectively. For LYG, screening was most effective earlier in life, at 55 years for a single colonoscopy, 49 and 64 years for two colonoscopies, 44, 57, and 69 years for three colonoscopies, and 40, 51, 61, and 72 years for four screening colonoscopies. The screening schedule was sensitive to adenoma detection rates during colonoscopies, individual adenoma risk, and adherence to screening. Among all CRC screening schedules optimized for LYG, only the four-colonoscopy schedule had an ICER above 100'000 dUSD/ dLYG. Our results largely endorse standard recommendations for CRC screening for the US population and the widely used "standard colonoscopy" screening scenario with colonoscopies at 50, 60 and 70 years provides a good balance of all investigated outcomes.

Several benefits can potentially be achieved by CRC screening. Some goals, such as reduction in incidence (preventing cancer) or reduction in mortality (preventing cancer-related death), are easily understood by patients. These goals have been the end points of randomized CRC screening studies [11,29,78–80] and form the basis of the guidelines of the American Cancer Society (ACS), the US Multi-Society Task Force on Colorectal Cancer and many more [34,81]. In contrast, LYG, as considered by the USPSTF, are possibly less intuitive to patients and cannot be directly measured in a clinical study since it cannot be known how long an individual would have lived without CRC. Hence, USPSTF recommendations were based on microsimulations instead [51,82,83]. Our study shows that the optimal screening schedule critically depends on the objective of screening. We thus suggest clear communication of screening objectives to practitioners and patients. Choosing LYG as an endpoint for health-economics studies has the unique advantage that interventions for different conditions can be compared. According to our results, three colonoscopies at 50, 60, and 70 years of age would

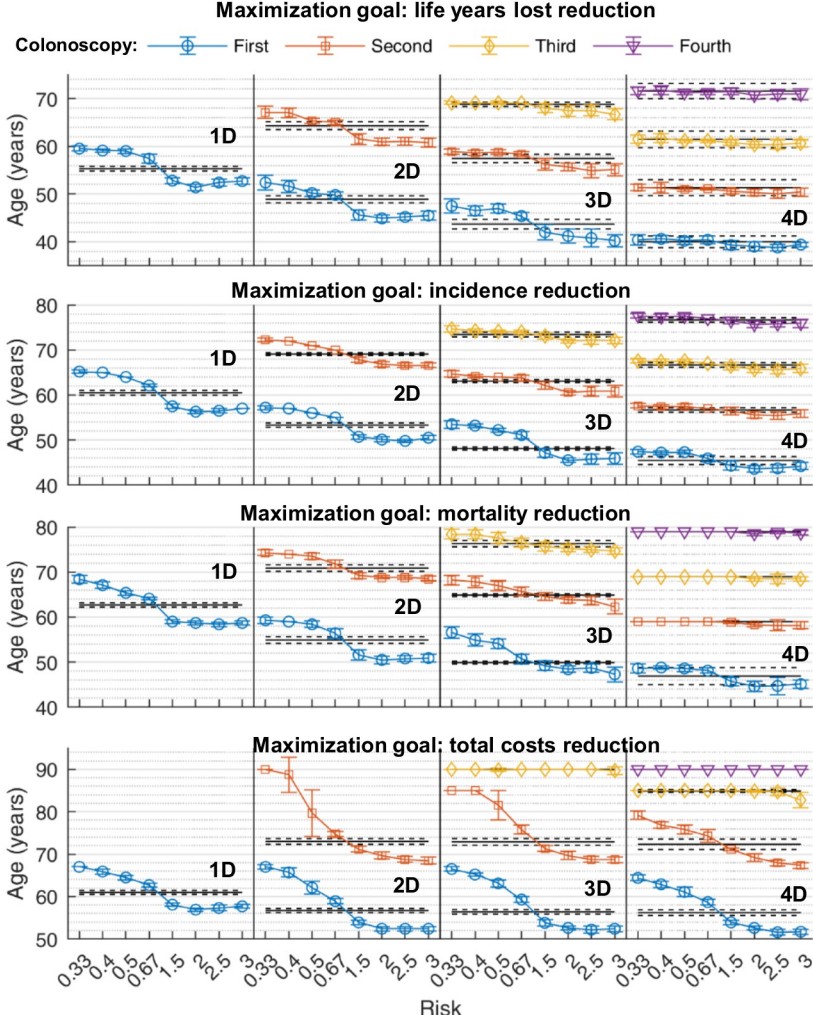

**Fig 5. Personalization of CRC screening.** We simulated a CRC screening test, which stratified individuals according to adenoma risk with the indicated risk ratios ranging from 0.33 to 3 as indicated. Optimal time points for one, two, three and four colonoscopies are indicated for an optimization of **(A)** reduction of life years lost, **(B)** CRC incidence reduction, **(C)** mortality reduction and **(D)** overall costs. Averages over N = 10 simulations on a population of 20 million individuals are indicated. Horizontal lines indicate timing of optimized screening schedules with standard deviation for the whole population.

yield a near-optimal incidence reduction, however, an optimum in the number of life years gained requires a start of screening approximately 5 years earlier.

In line with previous microsimulation studies [82,84], our results suggest an advantage of an early first examination in a three-colonoscopy schedule in US patients with an average risk for CRC. Compared to a standard screening schedule, an earlier start at the age of 44 would increase the number of LYG from 165.5 to 169.1 per 1000 patients (Table 1). This difference is likely due to a higher number of LYG by preventing or detecting CRC early in life compared to the older age group. Importantly, early screening has already been recommended by the ACS [81], resulting in considerable debate regarding cost-effectiveness [37,38,85]. Shifting screening to an earlier age indeed results in higher costs compared with standard screening. This is likely due to a slightly lower incidence reduction at early age (thus incurring more treatment costs) and a larger population at risk (resulting in more screening interventions).

Our results also indicate the effectiveness of four screening colonoscopies for the US population. Following some current recommendations [34,81], perfect compliance between 45 and 75 years of age would lead to four screening colonoscopies at 10-year time intervals. In our calculations, implementation of a fourth colonoscopy saved 10 additional life years per 1000 individuals and reduced CRC incidence by an additional 3%. The optimal schedule for four screening colonoscopies requires an initial colonoscopy at <50 years for optimal incidence and mortality reduction and at 40 years for an optimal number of life years gained. However, incremental benefits of the fourth colonoscopy are costly, as indicated by an ICER of >100'000 dUSD/dLYG and its implementation would depend on the willingness to pay by the society.

We used CMOSTv3 to perform a thorough sensitivity analyses for the optimal screening schedules. While CRC screening schedules were robust to changes in CRC survival, colonoscopy complication risk or cecal intubation rates, the timing of the first colonoscopy was sensitive to the efficiency of adenoma detection during colonoscopy with efficiencies favoring a later screening schedule. This finding suggests that detection of small adenomas during the first screening colonoscopy contributes to the effectiveness of screening. On the other hand, an early start of screening also seems to require the ability, commitment, and resources to perform colonoscopies with high quality [86]. Moreover, the timing of an optimal screening schedule was dependent on adenoma rates. Therefore, schedules adapted to the US would not perfectly apply to countries with lower adenoma rates and CRC incidence [87] and a later start of screening would be preferable.

The screening schedule was also sensitive to screening adherence. For all schedules, lower participation consistently favored a later start of the screening period and an earlier end thus resembling a N−1 scenario with high adherence. This caveat is important since screening participation ranges between 40% and 60% in the US and most industrialized countries [88,89]. However, adherence will likely be distributed unevenly in the population with low participation in some individuals and high adherence in others. Clear recommendations are difficult since adherence of an individual patient is only known retroactively and would have to be estimated from surrogate measures such as motivation to CRC screening, existing barriers to screening or adherence to other health recommendations. Therefore, in discussions with informed patients with lower ability or motivation for screening participation, the benefit of one or two screening colonoscopies between 55 and 69 years of age should be emphasized. In contrast, individuals with a predicted good adherence might benefit most from a third or fourth screening colonoscopy schedule and an early start of screening.

In another attempt to personalize screening recommendations, we stratified individuals into subgroups with different personal risks. According to our results, a higher personal risk would justify a start of screening earlier in life, whereas all remaining individuals should be screened later. Similarly, males should be screened earlier than females. Our observations might thus inform discussions of clinicians with patients who are not sure, whether they should receive a colonoscopy or wait. In the future, we envision a scenario, where epidemiologic, clinical, and genetic risk factors are used to calculate the most effective screening schedule.

Strengths of our study include our unbiased approach with a comprehensive search for an optimum, which was facilitated by the vastly improved performance of CMOSTv3. In contrast to previous microsimulation studies, which tested a limited choice of screening strategies, our results were obtained by testing the whole parameter space of colonoscopies from 20 to 90 years and may thus include schedules that were not considered in previous studies. We also performed a rigorous sensitivity analysis for factors that may affect the timing of the optimum and used open-source software that allows full reproduction of all results as well as exploration of other parameters in the future. Our study has various limitations: i) Some key parameters of

the natural history of CRC are still unknown. This is especially relevant for the adenoma dwell time, the time from the appearance of an adenoma until its transition to preclinical CRC. ii) The serrated adenoma pathway, which may account for up to 30% of CRC [90] and may be particularly relevant in the right colon, is not explicitly considered by our model. However, our model has implemented carcinoma directly developing from normal mucosa with a right-colonic preference that at least indirectly accounts for the serrated adenoma path. iii) Our study calculates life years but not quality adjusted life years. iv) The output metrices used as the goal for optimization was limited and the optimal schedule e.g., for maximizing colonoscopies per LYG should still be explored in future analyses. iv) Finally, microsimulation models cannot replace randomized studies. However, the search for the optimal time point of three colonoscopies required >4000 runs of CMOST, altogether simulating more than 80 billion patients. Considering these practical limitations, randomized studies comparing colonoscopy screening schedules are unlikely to be ever performed and microsimulation models might be the only means of approaching practical questions of CRC screening. The open-source model of CMOST uniquely allows the implementation of additional parameters and models. We are planning to further advance our software for broader usage and calculation of personalized screening as well as surveillance recommendations.

## Supporting information

**S1 Fig. Sensitivity analysis for a one-colonoscopy screening program.** Selected model parameters were varied by up to ±50%. Screening time points for a one-colonoscopy program are indicated for optimal CRC reduction of total costs. Dashed lines indicate the optimal schedule without parameter changes. Significance is tested using the Wilcoxon test against a distribution without parameter changes. $^*p < 0.05$, $^{**}p < 0.01$. $^{***}p < 0.001$. For all statistics N = 10 simulations were used with a population of 20 million individuals.
(PDF)

**S2 Fig. Sensitivity analysis for a two-colonoscopy screening program.** As Fig 1B, only optimal time points for a two-colonoscopy program are indicated with optimal CRC reduction of total costs.
(PDF)

**S3 Fig. Sensitivity analysis for a three-colonoscopy screening program.** As Fig 1B, only optimal time points for a three-colonoscopy program are indicated with optimal CRC reduction of total costs.
(PDF)

**S4 Fig. Sensitivity analysis for a four-colonoscopy screening program.** As Fig 1B, only optimal time points for a four-colonoscopy program are indicated with optimal CRC reduction of total costs.
(PDF)

**S1 Table. Benchmarks and results for CMOST calibration.**
(DOCX)

**S2 Table. Basic parameters for CMOST simulations.**
(DOCX)

**S3 Table. Input parameters for Mixed Integer Distributed Ant Colonoscopy Optimization (MIDACO).**
(DOCX)

## Author Contributions

**Conceptualization:** Niko Beerenwinkel, Amnon Sonnenberg, Benjamin Misselwitz.

**Data curation:** Niklas Krupka.

**Formal analysis:** Viktor Zaika, Meher K. Prakash, Benjamin Misselwitz, Jan Poleszczuk.

**Methodology:** Viktor Zaika, Meher K. Prakash, Benjamin Misselwitz, Jan Poleszczuk.

**Software:** Jan Poleszczuk.

**Writing – original draft:** Viktor Zaika, Niklas Krupka, Benjamin Misselwitz.

**Writing – review & editing:** Chih-Yuan Cheng, Michael Schlander, Brian M. Lang, Niko Beerenwinkel, Amnon Sonnenberg, Jan Poleszczuk.

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
