## [Decision Letter · Decision Letter 0]

16 Feb 2024

PONE-D-23-42000Optimal timing of a colonoscopy screening schedule depends on adenoma detection, adenoma risk, adherence to screening and the screening objective: a microsimulation studyPLOS ONE

Dear Dr. Misselwitz,

Thank you for submitting your manuscript to PLOS ONE. After careful consideration, we feel that it has merit but does not fully meet PLOS ONE’s publication criteria as it currently stands. Therefore, we invite you to submit a revised version of the manuscript that addresses the points raised during the review process.

We look forward to receiving your revised manuscript.

Kind regards,

Chih-Wei Tseng

Academic Editor

PLOS ONE

Journal Requirements:

"Swiss Cancer League grant No. KFS-5164-08-2020 Benjamin Misselwitz."

Reviewers' comments:

Reviewer's Responses to Questions

**Comments to the Author**

1. Is the manuscript technically sound, and do the data support the conclusions?

Reviewer #1: Yes

Reviewer #2: Partly

2. Has the statistical analysis been performed appropriately and rigorously? 

Reviewer #1: Yes

Reviewer #2: No

3. Have the authors made all data underlying the findings in their manuscript fully available?

Reviewer #1: No

Reviewer #2: No

4. Is the manuscript presented in an intelligible fashion and written in standard English?

Reviewer #1: No

Reviewer #2: Yes

5. Review Comments to the Author

Reviewer #1: The paper discusses the optimal offering of colonoscopy for the CRC eligible population. Authors have used a publicly available microsimulation model for their analysis. I have several comments that need to be addressed. Some of them are more critical for publication:

1- Importance of the study:

Authors mentioned several reasons that they have done the analysis. Some of them are very valid. There are other reasons that can be added: 1- colonoscopy backlog: In many countries (including US), although screening with routine COL is suggested, the capacity of colonoscopy is limited and patients may need to wait for a long time to get a COL. Your analysis showcase better use of COL, potentially removing any backlog barrier. 2- In some other countries, such as Thailand, the capacity of COL is much fewer than industrilized countries, and health policy makers need to use such analysis to determine the time and interval of screening. Authors can take a look to the following publicationL Wongseree, Peeradon, et al. "Dynamics of colorectal cancer screening in low and middle-income countries: A modeling analysis from Thailand." Preventive Medicine 175 (2023): 107694.

However, authors mentioned that CISNET only have done a few COL intervals and start-stop age. The reason that CISNET does that were not probably the resources. This is what USPSTF instructed them to do. USPSTF has its own criteria and reasoning behind when and why the screening should start from age 45 instead of 47. For instance, ease of interpretation of results and ease of use of the results by physicians and patients have taken into account. The implementation of the proposed optimal solution in national level is very complicated, thus hinder USPSTF, ACS, ACP, USMSTF to advocate for it.

2- Method:

Authors mentioned that they used CMOST model. It is expected to show the validity of the model compared to CISNET and other models in the US. Also external validation of the model, compared to clinical trials need to be shown. Here is an example of a model with validation points:

Vahdat, Vahab, et al. "Calibration and validation of the colorectal cancer and adenoma incidence and mortality (CRC-AIM) microsimulation model using deep neural networks." Medical Decision Making 43.6 (2023): 719-736.

Authors mentioned that "Due to the stochastic nature of the model, we performed 10 independent simulations for each set of evaluated model parameters". All simulation models are stochastic, but CISNET and others do not run the same model twice. I assume this is because the model is not using common random numbers (CRN). It is interesting for me to see that the model outcome fluctuate even with 20 million individuals (CISNET run their models with 2 million, as I recall).

The use of MIDACO is very vague to me. Authors are encouraged to describe how the simulation and optimization models are connected. Are you using the simulation outcomes as inputs to optimization model? Or the optimization model is imbedded into simulation model and change the inputs of simulation model, based on previous findings?

Authors mentioned: "The optimization problem supplied to MIDACO was a standard nonlinear minimization (minimize incidence/ mortality/ life years lost/ total costs)". My first impression was that the optimization model is handling all these objectives simultaneously, which then the normalization and weights of these functions need to be clarified. If these objectives are handled separately, then the balanced solutions need to be further described.

MIDACO seems to use PSO meta-heuristic. None of the metaheuristic algorithms may guarantee the optimal solution. How does authors deal with that?

Model inputs: the inputs to the model need to be clearly described and added as a table. These parameters include the costs, the adherence, adenoma detection rate, complication rates, and more.

3-Results:

Frankly, the results are not clearly described. I suggest working on clarity of results.

Fig 1.B: While I see a meaningful pattern of changes in ADR and risk in age of screening, the same pattern does not exist for adherence. My interpretation from Figue1 B is that no matter of adherence, if you want to minimize the LYG or incidence it will be always the same age. This is in contrast to the literature. If this is the case, authors need to clearly explain their thoughts around it.

4- Discussion:

I suggest authors to find one number for each screening interval and stick to it. Call out your number and let everyone knows what should be the exact time of screening if only two colonoscopy are offered.

We are not sue what USPSTF uses as their metric for choosing screening intervals. CISNET reports have LYG(benefit), cases averted(benefit), death averted (benefit), life-time colonoscopies (burden), complications (harms). CISNET are interested in benefit-to-burden ratios such as LYG per colonoscopies. It could be a good metric to optimize the model to. I refer authors the technical report of CISNET:

https://www.ncbi.nlm.nih.gov/books/NBK570833/

Reviewer #2: This is a paper which uses an existing microsimulation model, CMOST, to find optimal colonoscopy schedules for routine screening of asymptomatic individuals. It is a very interesting and useful topic, and one which has been generally underexplored. I commend the authors on tackling this, and providing a thoughtful analysis.

In general, I have concerns about the presentation of this paper, including how the modelling and statistical analysis are reported. The usefulness of this manuscript hinges on the reliability of the model, and understanding the precision and accuracy it can provide. In its current form, the manuscript does not convey to the reader any understanding of how accurate and/or precise the model is, how well it can replicate its targets, or any form of validation. I have noted below opportunities for the authors to add the required background; in the absence of these, the modelling is without foundation. There are some other minor points where the modelling is unclear; the use of the CHEERS checklist, or similar model evaluation, would help demonstrate the efficacy of their model and therefore the reliability of their results. There are also points where the statistical analysis and the results are reported in unclear ways; I have noted these below.

The authors should also acknowledge the work of Koffijberg et al, Using Metamodeling to Identify the Optimal Strategy for Colorectal Cancer Screening in Value in Health, 2020, which analyses optimal schedules for CRC via FIT screening.

Page 3/Introduction:

Line 11: Country-level colonoscopy screening programs are not common, to my knowledge; I believe that it is mostly used ad-hoc or through insurance (as in the US) rather than as part of a national program. Sources 30-33 support this – 30 is about the British surveillance program, not screening (i.e. not for asymptomatic individuals), and 31-33 are for US guidelines, which is not a screening “program” per se. This needs to be clarified in your text, with specific country references.

It is also worth noting that most country- or region-level screening programs for asymptomatic individuals are through FIT/FOBT (see e.g. Schreuders et al, Gut 2015). Many health-economic studies have shown that FIT is likely to be preferable vs colonoscopy for screening of an asymptomatic average-risk population (see e.g. Lew et al, International Journal of Cancer 2018, Zhong et al, Gastrointestinal Endoscopy 2019). However, there is no mention of FIT in your manuscript. Including this background, and the reasons colonoscopy would be used, would be very useful context for your manuscript.

Lines 30-38 could be clarified into a clearer aim. Much of what is included is written as methods and results, and should be moved to those sections as appropriate.

Page 4/Methods:

Line 4: It is unclear what re-engineered means – do you just mean you reimplemented the algorithms, or were they redesigned? If the algorithms have been changed, the differences from what has already been published should be described in detail. Otherwise please clarify the wording, as re-engineer implies a redesign of the model.

Line 6: As this includes a new set of calibration targets, these targets should be included in the manuscript (potentially in the supplementary materials), as well as the model performance on these calibration targets. As the meaningfulness of your manuscript hinges on the reliability of this model, this is critical. I also note that in reference 35, the targets of the CMOST model are included as table S1 but it is not clear whether the model is able to successfully reproduce these targets. This should be included in this manuscript.

The SEER data only covers the US population, and therefore is most relevant to that population. As other countries can have dramatically different colorectal cancer incidence, mortality, and prevalence rates, you should make it clear to the reader throughout the manuscript, including in the introduction, that your baseline modelling is for the US population, though other populations could be approximated by the sensitivity analyses.

Line 13: It is very strange that your model includes CRCs with no precursor states. In Prakash et al 2017, it is stated that “this direct cancer pathway represents cancers with precursors difficult or impossible to detect by colonoscopy”. However, you are allegedly simulating a patient’s actual cancer/adenoma state (as evidenced by the simulation of screening, and the associated detection rates) rather than their observable (and potentially incorrect) state. This explanation is therefore a bit unsatisfactory. It seems to me that you are using this transition to “make up for” discrepancies between the data and your model performance. Perhaps this has to do with the timestep? A further explanation of this would be useful, as in the current manuscript it may appear that you are asserting that CRC can form spontaneously with no precursor, which I don’t think is the intention.

Line 18: Specify US guidelines.

Line 22: Is the three-month timestep a change from previous versions of CMOST? If so please make this explicit.

Lines 29-32: Were these 10 simulations on the same parameter set? If so, it is unclear to me what the benefit of running these simulations, or of running the test afterwards is, vs combining the simulation outputs to increase the statistical power of your simulation (in these contexts, 20 million simulated individuals may not in fact be particularly “large”). Are these ten simulations the methods by which you generate the standard deviations included below? If so I don’t think these are meaningful standard deviations, as they are driven entirely by your choice of simulation size rather than anything inherent to the uncertainty of your model or the size of any population of interest.

Line 35-36: For each value required by the solver, was the model re-run, or was this a metamodel-type analysis like that in Koffijberg et al, Value in Health 2020?

Page 5

Line 14: What economic perspective was used? What costs are included?

Line 28: This should be rewritten to make it clear that there is no fixed “current practice” for discounting, with various discount rates used based on setting and context. In particular the cost-effectiveness of screening programmes can be highly dependent on the choice of discount rate. A specific reference should be cited.

Line 36: Each parameter was changed separately, correct? If so this should be specified as a one-way sensitivity analysis.

Line 36: Is the baseline adherence 100%?

Page 6

Lines 10-11: Based on my understanding, both the proportion and the relative risk of this “high-risk” group is entirely hypothetical and not based on actual groups within the population – is this correct? This should be made clearer, though the analysis is still very valuable.

Page 8

Table 1:

This should be referenced in the text earlier.

The asterisks do not seem to be referencing anything.

“Standard screening” has not been defined in the text, as far as I can tell.

What are these standard deviations based on?

What are these % changes vs? No screening? If so, the change in life-years gained is very strange. It is also not clear what the number in brackets for LYG is.

The mortality reduction from a single colonoscopy also seems very high. Some reference to existing literature/validation of this rate would be useful context.

Page 9

Discussion:

It should be noted that the “standard screening” scenario in Table 1 does in fact provide a good balance between the outcomes, based on the numbers you present; a partial endorsement of the current recommendations with scope to improve in line with your other findings.

Line 7: The “high risk”/”low risk” designators do not make sense in the context they are presented, where the “high risk” group can have lower cancer rates than the low-risk group in Figure 5. I suggest you revise the wording around this analysis; perhaps just designate it as a “subgroup with a different risk”. The low risk also seems to be “average” or “general population” risk. This needs significant clarification.

Line 26: It is strange to assert that LYG are “less intuitive” – you would be better off expanding on why they are less practical to measure.

A limitation of your study is that you have used life-years, rather than quality-adjusted life years, as a measure; this should be mentioned and the difference noted (especially when considering high-risk groups).

Page 10:

Line 13: Unclear what “high performance” means here.

Line 22: This first sentence is incomplete.

Line 29: Although this is a strong conclusion, “recommending early start for those with strong adherence” is a meaningless recommendation in practice – you can only know this retroactively! Suggest removing or recontextualising.

Good suggestion of moving towards personalised recommendations; you could argue that your findings would be useful to inform ad-hoc recommendations for clinicians to inform people who are not sure if they should receive a colonoscopy or wait.

Figure 1: Suggest moving B to supplementary materials, or finding a simpler way to convey this.

Figure 2: A) I do not understand the advantage of including the non-dominated solutions – the colours are sufficiently clear. “Effectiveness” is misspelled. Again B is not sufficiently clear for inclusion in the main text, in my opinion.

Figure 3: Presenting this as a 3D figure has no advantage over presenting it as three 2D figures, side by side, and makes it more difficult to read. It is not possible for a reader to visually interpret the line indicating the optimum on the z-axis. Again, omitting the non-dominated solutions will improve readability, and figure B should be moved to supplementary material.

Figure 4: Unclear to me how this is an “ICER analysis”; the outcome listed is the discounted LYG? The ICER is not shown on this figure, as far as I can tell.

Figure 5: As noted in the text, “high” and “low” risk are very misleading when the low-risk group can have risk ratios less than 1.

6. PLOS authors have the option to publish the peer review history of their article (what does this mean?). If published, this will include your full peer review and any attached files.

Reviewer #1: No

Reviewer #2: No

---

## [Author Response · Author response to Decision Letter 0]

3 Apr 2024

Journal Requirements: When submitting your revision, we need you to address these additional requirements.

Response: this has been addressed. 

"Swiss Cancer League grant No. KFS-5164-08-2020 Benjamin Misselwitz."

Response: this has been addressed. 

 Reviewer #1: No

Reviewer #2: No

Response: We respectfully disagree. The code is fully available, and a link has been provided within the manuscript. The code can be used to reproduce all results from our manuscript. 

5. Review Comments to the Author

Reviewer #1: The paper discusses the optimal offering of colonoscopy for the CRC eligible population. Authors have used a publicly available microsimulation model for their analysis. I have several comments that need to be addressed. Some of them are more critical for publication:

1- Importance of the study:

Authors mentioned several reasons that they have done the analysis. Some of them are very valid. There are other reasons that can be added: 1- colonoscopy backlog: In many countries (including US), although screening with routine COL is suggested, the capacity of colonoscopy is limited and patients may need to wait for a long time to get a COL. Your analysis showcase better use of COL, potentially removing any backlog barrier. 2- In some other countries, such as Thailand, the capacity of COL is much fewer than industrilized countries, and health policy makers need to use such analysis to determine the time and interval of screening. Authors can take a look to the following publication L Wongseree, Peeradon, et al. "Dynamics of colorectal cancer screening in low and middle-income countries: A modeling analysis from Thailand." Preventive Medicine 175 (2023): 107694.

Response: We agree with the ideas of the reviewer and modified the second paragraph of the introduction with the suggested reference (1) (page 3, line 15).

However, authors mentioned that CISNET only have done a few COL intervals and start-stop age. The reason that CISNET does that were not probably the resources. This is what USPSTF instructed them to do. USPSTF has its own criteria and reasoning behind when and why the screening should start from age 45 instead of 47. For instance, ease of interpretation of results and ease of use of the results by physicians and patients have taken into account. The implementation of the proposed optimal solution in national level is very complicated, thus hinder USPSTF, ACS, ACP, USMSTF to advocate for it.

Response: We would like to thank the reviewer for these comments. The reasoning for the CISNET group not to test yearly intervals is now better explained (page 3, line 33).

2- Method:

Authors mentioned that they used CMOST model. It is expected to show the validity of the model compared to CISNET and other models in the US. Also external validation of the model, compared to clinical trials need to be shown. Here is an example of a model with validation points:

Vahdat, Vahab, et al. "Calibration and validation of the colorectal cancer and adenoma incidence and mortality (CRC-AIM) microsimulation model using deep neural networks." Medical Decision Making 43.6 (2023): 719-736.

Response: We agree with the reviewer, that validation is crucial for any modelling study, and we read with great interest the recent study by Vahdat et. al. (2) (now quoted as new reference 44). However, we would like to respectfully note that CMOST validation had indeed been performed in the Paper by Prakash et al. (3), following a highly similar approach as used by Vahdat et al. (2). In brief, we performed a careful comparison of representative CMOST results with outcomes of CISNET microsimulation models (Figure 4 of the study by Prakash et al.). In addition, external validation had been performed, using data from the study by Schoen et al. (4), (compare Table 3, Effects of rectosigmoidoscopy screening for CRC prevention combined with 11.9-year follow-up from the study by Prakash et al.). Our results show that predictions by CMOST are in line with predictions of the CISNET model as well as clinical data. 

However, we agree with the reviewer that this needs to be better explained in our manuscript. We now provide this relevant information in our manuscript (page 5, line 13). Further, benchmarks and results of our model calibrations are now provided (compare new Supplementary Table 1). 

Authors mentioned that "Due to the stochastic nature of the model, we performed 10 independent simulations for each set of evaluated model parameters". All simulation models are stochastic, but CISNET and others do not run the same model twice. I assume this is because the model is not using common random numbers (CRN). It is interesting for me to see that the model outcome fluctuate even with 20 million individuals (CISNET run their models with 2 million, as I recall).

Response: We appreciate the insightful comments of this reviewer. Our model indeed does not use common random numbers. We rather use a set of 10 seed numbers as input parameters for the Mercenne-Twister uniform pseudo-random number generator. 

We agree with the reviewer that some numbers in our results fluctuate, but not for simple outcomes of CMOST: the standard deviation is indeed quite small for incidence or mortality reduction due to colonoscopy (compare confidence intervals in Table 1). However, our numbers fluctuate for the dependencies of these results on model assumptions (e.g., Figures 1B, 2B, 3B), which represent complex calculations. One needs to remember that during optimization checking any new set of parameters requires stochastic simulation – this is another source of noise as there is non-zero chance that comparison between two sets of parameters will be different depending on the random seed value. To address this error, we consistently provide confidence intervals or other error measures. We now acknowledge and explain this fluctuation better in the new version of the manuscript (page 5, line 5). 

The use of MIDACO is very vague to me. Authors are encouraged to describe how the simulation and optimization models are connected. Are you using the simulation outcomes as inputs to optimization model? Or the optimization model is imbedded into simulation model and change the inputs of simulation model, based on previous findings?

Authors mentioned: "The optimization problem supplied to MIDACO was a standard nonlinear minimization (minimize incidence/ mortality/ life years lost/ total costs)". My first impression was that the optimization model is handling all these objectives simultaneously, which then the normalization and weights of these functions need to be clarified. If these objectives are handled separately, then the balanced solutions need to be further described.

MIDACO seems to use PSO meta-heuristic. None of the metaheuristic algorithms may guarantee the optimal solution. How does authors deal with that?

Response: We want to apologise for the vague description of the optimization method. We have thoroughly reworked the section (page 5, lines 14-24) We are aware that none of the metaheuristic algorithms may guarantee optimal solution and that is exactly why in Figs. 1 – 3 we plot the landscape of the optimization problem to see if the resulting set of optimal parameter falls within the compact areas. 

Model inputs: the inputs to the model need to be clearly described and added as a table. These parameters include the costs, the adherence, adenoma detection rate, complication rates, and more.

Response: We agree with the reviewer, the model inputs are now summarized in the new Supplementary Table 1.

3-Results:

Frankly, the results are not clearly described. I suggest working on clarity of results.

Fig 1.B: While I see a meaningful pattern of changes in ADR and risk in age of screening, the same pattern does not exist for adherence. My interpretation from Figue1 B is that no matter of adherence, if you want to minimize the LYG or incidence it will be always the same age. This is in contrast to the literature. If this is the case, authors need to clearly explain their thoughts around it.

Response: We agree with the reviewer, the results of Figure 1B should be described clearer and we are grateful to the reviewer for pointing this out. The optimal year for a single colonoscopy remains the same for individuals with high adherence as well as low adherence. This is due to the fact that for a screening program with a single colonoscopy (as assumed in Figure 1), a lower adherence simply results in non-participation of a fraction of the population in the program. This is now much better explained in the results explaining the findings in Figure 1B (page 7, line 20). We also show that for multiple colonoscopies adherence significantly impacts the optimal age of screening (see Fig 2B and Fig 3B). To increase clarity, we now provide reading examples in the legends to Figures 1B, 2B and 3B.

4- Discussion:

I suggest authors to find one number for each screening interval and stick to it. Call out your number and let everyone knows what should be the exact time of screening if only two colonoscopy are offered.

Response: We agree, our findings including the calculated optimal years for a 1-, 2- 3- and 4-colonoscopy schedule are now re-emphasized in the first paragraph of the discussion. 

We are not sue what USPSTF uses as their metric for choosing screening intervals. CISNET reports have LYG(benefit), cases averted(benefit), death averted (benefit), life-time colonoscopies (burden), complications (harms). CISNET are interested in benefit-to-burden ratios such as LYG per colonoscopies. It could be a good metric to optimize the model to. I refer authors the technical report of CISNET:

https://www.ncbi.nlm.nih.gov/books/NBK570833/

Response: We agree with the reviewer. We added colonoscopies per LYG as well as colonoscopies per case prevented, as suggested by the reviewer to Table 1. We agree that LYG per colonoscopy would be an interesting metric for optimization, but in our opinion out of scope for the current study. This is now mentioned in the limitation sections (page 12, line 35). 

Reviewer #2: This is a paper which uses an existing microsimulation model, CMOST, to find optimal colonoscopy schedules for routine screening of asymptomatic individuals. It is a very interesting and useful topic, and one which has been generally underexplored. I commend the authors on tackling this, and providing a thoughtful analysis.

Response: We are grateful to the reviewer for the general positive evaluation of our work. 

In general, I have concerns about the presentation of this paper, including how the modelling and statistical analysis are reported. The usefulness of this manuscript hinges on the reliability of the model, and understanding the precision and accuracy it can provide. In its current form, the manuscript does not convey to the reader any understanding of how accurate and/or precise the model is, how well it can replicate its targets, or any form of validation. I have noted below opportunities for the authors to add the required background; in the absence of these, the modelling is without foundation. There are some other minor points where the modelling is unclear; the use of the CHEERS checklist, or similar model evaluation, would help demonstrate the efficacy of their model and therefore the reliability of their results. There are also points where the statistical analysis and the results are reported in unclear ways; I have noted these below.

The authors should also acknowledge the work of Koffijberg et al, Using Metamodeling to Identify the Optimal Strategy for Colorectal Cancer Screening in Value in Health, 2020, which analyses optimal schedules for CRC via FIT screening.

Response: We agree and added the reference to the paper (new reference 41, page 3, line 21). 

Page 3/Introduction:

Line 11: Country-level colonoscopy screening programs are not common, to my knowledge; I believe that it is mostly used ad-hoc or through insurance (as in the US) rather than as part of a national program. Sources 30-33 support this – 30 is about the British surveillance program, not screening (i.e. not for asymptomatic individuals), and 31-33 are for US guidelines, which is not a screening “program” per se. This needs to be clarified in your text, with specific country references.

Response: We agree with the reviewer, there are not many country-based screening programs, and we now introduce CRC screening in more general terms without mentioning a program (page 3, lines 13-14). Previous reference 30 was removed. 

It is also worth noting that most country- or region-level screening programs for asymptomatic individuals are through FIT/FOBT (see e.g. Schreuders et al, Gut 2015). Many health-economic studies have shown that FIT is likely to be preferable vs colonoscopy for screening of an asymptomatic average-risk population (see e.g. Lew et al, International Journal of Cancer 2018, Zhong et al, Gastrointestinal Endoscopy 2019). However, there is no mention of FIT in your manuscript. Including this background, and the reasons colonoscopy would be used, would be very useful context for your manuscript.

Response: We feel this is an important criticism and the first 3 paragraphs of the introduction have been rewritten and now introduce FIT as a highly cost-effective option; colonoscopy and FIT are subsequently discussed in a balanced way (page 3, line 6, 15) However, we feel that a discussion of the benefits of FIT vs. colonoscopy in even more detail would be out of scope for this paper. 

Lines 30-38 could be clarified into a clearer aim. Much of what is included is written as methods and results, and should be moved to those sections as appropriate.

Response: We agree and shorted and rephrased the last paragraph of the introduction to make the aims of our study much clearer (page 3, line 37 until page 4, line 3). 

Page 4/Methods:

Line 4: It is unclear what re-engineered means – do you just mean you reimplemented the algorithms, or were they redesigned? If the algorithms have been changed, the differences from what has already been published should be described in detail. Otherwise please clarify the wording, as re-engineer implies a redesign of the model.

Response: We agree with the reviewer and apologize for the unclear wording. The model has been re-implemented in C++ for much faster speed, while the logic and assumptions of the model have not been changed. This is now written much clearer in the new version of the manuscript (page 4. line 8). 

Line 6: As this includes a new set of calibration targets, these targets should be included in the manuscript (potentially in the supplementary materials), as well as the model performance on these calibration targets. As the meaningfulness of your manuscript hinges on the reliability of this

---

## [Decision Letter · Decision Letter 1]

12 Apr 2024

PONE-D-23-42000R1Optimal timing of a colonoscopy screening schedule depends on adenoma detection, adenoma risk, adherence to screening and the screening objective: a microsimulation studyPLOS ONE

Dear Dr. Misselwitz,

Thank you for submitting your manuscript to PLOS ONE. After careful consideration, we feel that it has merit but does not fully meet PLOS ONE’s publication criteria as it currently stands. Therefore, we invite you to submit a revised version of the manuscript that addresses the points raised during the review process.

We look forward to receiving your revised manuscript.

Kind regards,

Chih-Wei Tseng

Academic Editor

PLOS ONE

Journal Requirements:

**Additional Editor Comments:**

Please ensure that your manuscript meets PLOS ONE's style requirements, including those for file naming. For example, the simple summary section is not necessary

Reviewers' comments:

Reviewer's Responses to Questions

**Comments to the Author**

1. If the authors have adequately addressed your comments raised in a previous round of review and you feel that this manuscript is now acceptable for publication, you may indicate that here to bypass the “Comments to the Author” section, enter your conflict of interest statement in the “Confidential to Editor” section, and submit your "Accept" recommendation.

Reviewer #2: (No Response)

2. Is the manuscript technically sound, and do the data support the conclusions?

Reviewer #2: Yes

3. Has the statistical analysis been performed appropriately and rigorously? 

Reviewer #2: Yes

4. Have the authors made all data underlying the findings in their manuscript fully available?

Reviewer #2: (No Response)

5. Is the manuscript presented in an intelligible fashion and written in standard English?

Reviewer #2: Yes

6. Review Comments to the Author

Reviewer #2: Most of my concerns have been well-addressed, especially with the inclusion of Supplementary Table 1. As this table shows that the calibration of the model is good, but perhaps not perfect, I would appreciate if the authors could mention the quality of the calibration in the results or discussion. The authors should be able to quantify the discrepencies between the targets and the model, and provide commentary on how accurate the model is and how useful the findings based on it are (I personally think it would be very reliable).

Table 1 - The "Life-years gained" outcome still needs to be rewritten if the percentage is your main outcome. Unless I am misunderstanding, you mean "Reduction in life-years lost to CRC death" - correct? The distinction is very important to make the comparator clear, as otherwise it appears as "Number of life-years in the total US population".

These small changes would improve the manuscript. Otherwise, I strongly recommend it for publication as a useful and fascinating addition to the literature on colonoscopy screening.

7. PLOS authors have the option to publish the peer review history of their article (what does this mean?). If published, this will include your full peer review and any attached files.

Reviewer #2: No

---

## [Author Response · Author response to Decision Letter 1]

6 May 2024

Journal Requirements:

Response: The references have not been modified. 

Additional Editor Comments:

Please ensure that your manuscript meets PLOS ONE's style requirements, including those for file naming. For example, the simple summary section is not necessary

Response: We agree, we double checked the style requirements. The summary section has now been deleted (page 2).

Comments to the Author

6. Review Comments to the Author

Reviewer #2: Most of my concerns have been well-addressed, especially with the inclusion of Supplementary Table 1. As this table shows that the calibration of the model is good, but perhaps not perfect, I would appreciate if the authors could mention the quality of the calibration in the results or discussion. The authors should be able to quantify the discrepencies between the targets and the model, and provide commentary on how accurate the model is and how useful the findings based on it are (I personally think it would be very reliable).

Response: We agree and commented on the quality of the fit and possible consequences for the reliability of our mode (pages 4-5, lines 33-5).

Table 1 - The "Life-years gained" outcome still needs to be rewritten if the percentage is your main outcome. Unless I am misunderstanding, you mean "Reduction in life-years lost to CRC death" - correct? The distinction is very important to make the comparator clear, as otherwise it appears as "Number of life-years in the total US population".

Response: We agree and modified the text accordingly. Life years gained is now appropriately defined and we also provide the definition in the headings of Table 1 (Table 1, page 10, line 4). 

These small changes would improve the manuscript. Otherwise, I strongly recommend it for publication as a useful and fascinating addition to the literature on colonoscopy screening.

Response: We would like to thank this reviewer for his efforts during this very constructive reviewing process and the general appreciation of our work.

---

## [Editor Report · Decision Letter 2]

13 May 2024

Optimal timing of a colonoscopy screening schedule depends on adenoma detection, adenoma risk, adherence to screening and the screening objective: a microsimulation study

PONE-D-23-42000R2

Dear Dr. Misselwitz,

We’re pleased to inform you that your manuscript has been judged scientifically suitable for publication and will be formally accepted for publication once it meets all outstanding technical requirements.

Kind regards,

Chih-Wei Tseng

Academic Editor

PLOS ONE
---

## [Editor Report · Acceptance letter]

16 May 2024

PONE-D-23-42000R2 

PLOS ONE

Dear Dr. Misselwitz, 

I'm pleased to inform you that your manuscript has been deemed suitable for publication in PLOS ONE. Congratulations! Your manuscript is now being handed over to our production team.

Kind regards, 

on behalf of

Dr. Chih-Wei Tseng 

Academic Editor

PLOS ONE